# Effect of Length of Cellulose Nanofibers on Mechanical Reinforcement of Polyvinyl Alcohol

**DOI:** 10.3390/polym14010128

**Published:** 2021-12-30

**Authors:** Mengxia Wang, Xiaran Miao, Hui Li, Chunhai Chen

**Affiliations:** 1Center for Advanced Low-Dimension Materials, State Key Laboratory for Modification of Chemical Fibers and Polymer Materials, College of Material Science and Engineering, Donghua University, Shanghai 201620, China; 2180257@mail.dhu.edu.cn (M.W.); cch2018@jlu.edu.cn (C.C.); 2Shanghai Synchrotron Radiation Facility, Shanghai Advanced Research Institute, Chinese Academy of Sciences, Shanghai 201204, China; 3Shanghai Institute of Applied Physics, Chinese Academy of Sciences, Shanghai 201800, China

**Keywords:** TEMPO-mediated oxidized cellulose nanofibrils, morphology and dimension, reinforcement, polyvinyl alcohol

## Abstract

Cellulose nanofibers (CNF), representing the nano-structured cellulose, have attained an extensive research attention due to their sustainability, biodegradability, nanoscale dimensions, large surface area, unique optical and mechanical performance, etc. Different lengths of CNF can lead to different extents of entanglements or network-like structures through van der Waals forces. In this study, a series of polyvinyl alcohol (PVA) composite films, reinforced with CNF of different lengths, were fabricated via conventional solvent casting technique. CNF were extracted from jute fibers by tuning the dosage of sodium hypochlorite during the TEMPO-mediated oxidation. The mechanical properties and thermal behavior were observed to be significantly improved, while the optical transparency decreased slightly (Tr. > 75%). Interestingly, the PVA/CNF20 nanocomposite films exhibited higher tensile strength of 34.22 MPa at 2 wt% filler loading than the PVA/CNF10 (32.55 MPa) while displayed higher elastic modulus of 482.75 MPa than the PVA/CNF20 films (405.80 MPa). Overall, the findings reported in this study provide a novel, simple and inexpensive approach for preparing the high-performance polymer nanocomposites with tunable mechanical properties, reinforced with an abundant and renewable material.

## 1. Introduction

Nanotechnology has impacted the wide areas such as medicine, electronics and food technology by manipulating the nanomaterials for various purposes [1,2,3,4,5]. The material development has significant improved owing to the incorporation of nanotechnology. In recent years, the use of nanomaterials as nano-reinforcement in the polymer composites has been widely reported [6,7]. The nanoscale fibers used for reinforcing the polymer matrices represent an important class of nano-reinforcements [8,9,10]. The fibers are dispersed at nanoscale, thus, a significantly less filler fraction is needed as compared to the conventional reinforcing materials, while the properties of composites are markedly improved [11,12]. To minimize the environmental health and safety concerns, the sustainable and environmentally friendly development of nanomaterials is highly desired [13,14]. One of the most promising strategies for the sustainable development involving nanotechnology is the isolation of nanomaterials from low-cost, abundant, renewable and biodegradable resources [15]. Thus, the green nanomaterials have received major attention recently, thus, enhancing the development of natural and renewable products [16,17].

Nanocellulose possesses valuable characteristics such as a renewability and biodegradability with satisfactory mechanical properties when incorporated in a polymer matrix [18,19,20]. Owing to the depletion of the raw materials to develop the synthetic polymers, the use of nanocellulose as biopolymer reinforcing agent can be promising [21,22,23,24]. The literature studies reveal that research interest in the nanocomposites consisting of nanocellulose and polymer matrices has significantly enhanced in recent years. The reported increment in the tensile properties, crystallinity index and specific surface area of the composites successfully demonstrate the potential use of nanocellulose as reinforcement for developing the polymer nanocomposites [25,26,27,28]. Overall, nanocellulose represents a superior eco-friendly material as compared with the inorganic nanoscale materials used as reinforcements [29,30]. However, a few challenges exist with respect to the use of nanocellulose in polymers for achieving good performance, including the efficient dispersion of particles in the matrix, compatibility of nano-reinforcement in the matrix and development of suitable methods for the material processing, which also determining the final properties of the cellulose nanocomposites [21,24,25].

With respect to the mechanical properties of the nanocellulose-based nanocomposites, regardless of the type of polymer matrix or processing method, the addition of nanocellulose as a reinforcement phase in the polymer matrices can improve the mechanical performance of the resulting materials. A number of studies have reported that the addition of nanocellulose in the polymer matrices leads to an enhanced tensile modulus of the materials, which is attributed to the geometry and stiffness of nanocellulose [31,32,33]. In addition, the enhancement in the mechanical properties also corresponds to the strong interactions between the matrix and nanocellulose, which restrict the movement of the polymer chains, thereby hindering its deformation [22,34]. Apart from mechanical properties, several studies have shown that the incorporation of nanocellulose can also improve the thermal properties of the materials [35,36,37]. Generally, the thermal stability of the polymer nanocomposites has been reported to increase with the incorporation of nanocellulose, which is attributed to the restricted mobility of the polymer chains due to the addition of nanocellulose as well as the homogeneous distribution of nanocellulose in the polymer matrix [22,24,34,38].

Cellulose nanofibrils with different size presents different surface charge, dimension, morphology, and crystal structure, which precisely make CNF-based materials possess distinct properties and performances. Yang et al. produced cellulose nanofibrils with different lengths by the high-pressure homogenization and the results showed that the short cellulose nanofibril exhibited higher hydrophobicity and low interfacial tension [31]. Isogai et al. found self-standing films manufactured by longer cellulose nanofibrils displayed obviously higher tensile strengths, elongations at break and crystallinity indexes [32]. Besides, Wu et al. proposed that cellulose fibril with large size had higher ice recrystallization inhibition (IRI) activity [33]. These typical results apparently demonstrated that composites enhancing by cellulose nanofibril with various sizes could showcase significantly different properties as well. Moreover, polyvinyl alcohol (PVA), a water-soluble crystalline polymer, has extensive prospects in the fields of food packaging film materials, textile, artificial organs and medical gels biomedical fields, etc, which predominantly put down to its biodegradability, biocompatibility, excellent film-forming properties and gas barrier properties [39,40,41]. PVA is normally used composites to lower down the cost of product without any compromise or partial compromise with its properties and the hydrophilic composition, which consists of hydroxyl groups in the PVA structure, is well-linked with carbohydrates, creating significant agreement in composites [40,42,43].

To the best of our knowledge, the previous studies have primarily concentrated on the impact of nanocellulose on the optical transparency, morphological properties and mechanical properties of the nanocomposites, and the effect of the CNF length at nanoscale on the mechanical properties has been rarely studied. In this study, three kinds of CNF with different dimensions were isolated from jute fibers by tuning the dosage of sodium hypochlorite during the TEMPO-mediated oxidation, and the CNF reinforced PVA composites were fabricated via the conventional solvent casting/evaporation technique. The surface charge, morphology, physical properties and crystallinity index of CNF were successively characterized, and the impact of CNF length as well as the evolution of the multi-level microstructure in the nanocellulose-based composites during stretching were studied to explore the mechanism of reinforcement by nanocellulose, so as to provide an effective reference for the subsequent preparation of nano-matrix composites with higher performance.

## 2. Materials and Methods

### 2.1. Materials

The pristine jute fibers were provided by Redbud Textile Tech. Inc., Suzhou, China. The fibers were sufficiently dried in a vacuum drying oven at 70 °C for over 24 h. Polyvinyl alcohol with a degree of alcoholysis of 97.0–98.8 mol% was procured from Shanghai Yuanye Biological Technology Co. Ltd. (Shanghai, China) 2,2,6,6-tetrmethylpiperidine-1-oxyl radical (TEMPO, 98%), sodium bromide (NaBr, 99.6%) and sodium hypochlorite (NaClO, 6–14%) were supplied by Shanghai Macklin Biochemical Co. Ltd. (Shanghai, China) and were used as received. The sodium hydroxide (NaOH, ≥96.0%) and ethanol were purchased from Shanghai Titan Scientific Co. Ltd. and utilized without any further purification. All reagents were of analytical grade, and ultrapure water generated using the Milli-Q plus water purification system (Millipore Corporation, Billerica, MA, USA) was used.

### 2.2. Preparation of CNF of Different Sizes

The jute fibers were alkali treated by using 3 wt% NaOH solution at 70 °C for 4 to remove a majority of lignin, hemicellulose and pectin. Subsequently, the pre-treated jute fibers were filtered and washed for several times with deionized water, followed by drying in a vacuum oven at 80 °C for 6 h. The preparation of CNF by the TEMPO-mediated catalytic oxidation has been reported in detail by Isogai et al. [44,45]. Briefly, 1 g alkalized jute fibers were dispersed in 100 mL deionized water. The suspension was stirred for 1 h, followed by the addition of NaBr (0.33 g) and TEMPO (0.033 g). It is particularly emphasized that the morphology and size of CNF can be changed by adding different contents of NaClO into the system, which is also the most important point different from other studies [29,30]. The NaClO solutions (20 g/10 g/5 g) were subsequently added to the suspension to initiate the reaction at room temperature. The pH of the mixture was maintained between 10.6 and 10.8 by adding 1 wt% NaOH solution. The reaction was terminated by adding ethanol (7.0 mL) until the pH of the system no longer changed. The translucent jelly-like CNF were obtained after repeated centrifugal washing, which were denoted as CNF20, CNF10 and CNF5, respectively. Later, the resulting CNF were stored at 4 °C without any treatment for subsequent utilization.

### 2.3. Preparation of PVA/CNF Films

The PVA/CNF composite films were fabricated by using the solvent casting technique (Figure 1). PVA was first heated to dissolve in deionized water to obtain 8 wt% solution and were subsequently mixed with CNF dispersion in different ratios in a flask under vigorous stirring for 1.5 h to obtain the PVA/CNF hybrid solutions. Here, the weight content of CNF in the CNF/PVA nanocomposite solutions was varied from 0 to 2 wt% (based on the PVA solid weight). The mixed solutions were diluted to a solid content of 4 wt%, and the same quantity of the mixed solutions was subsequently poured in the TFE molds with the same diameter and dried for 48 h at 25 °C, 1% humidity. Eventually, a series of nanocomposites with different CNF contents was obtained.

### 2.4. Transmission Electron Microscopy (TEM)

The CNF morphology was analyzed by using a transmission electron microscope (JEM-2100F, Japan JEOL Ltd, Beijing, China) equipped with a Gatan 1 k × 1 k CCD camera. About 10 μL CNF ethanol suspensions were deposited onto glow-discharged carbon-coated TEM grids.

### 2.5. Fourier Transform Infrared Spectroscopy (FTIR)

The FTIR spectra of polyvinyl alcohol, nanocellulose and the reinforced composites were acquired by using a Bruker Tensor II infrared spectrometer (Karlsruhe, Germany) in the range 400–4000 cm^−1^.

### 2.6. Particle Size and Zeta Potential Measurement

Zetasizer Nano ZS (Malvern Instruments Ltd., Malvern, UK) was used to measure the surface charge and estimate the size of the nanocellulose particles in aqueous suspension. The suspensions were diluted 1000 times and sonicated for 30 min. The measurements were replicated 3 times, and an average value was presented. Additionally, the zeta potential distribution of the CNF suspensions was monitored with no ionic strength adjustment.

### 2.7. Mechanical Properties

The tensile analysis of the PVA film and nanocellulose reinforced PVA composites were carried out using an INSTRON 5966 electronic universal material testing machine (Boston, America), equipped with a 10 kN load cell. The film specimens were cut in a rectangular shape with dimensions 40 × 5 mm and a thickness of 0.2~0.3 mm. The tensile measurements were conducted using a head speed of 6 mm min^−1^ and a span length of 10 mm. Each membrane was tested at least 5 times under same conditions, and the average value were reported.

### 2.8. Thermal Properties

The thermal properties of the PVA/CNF composite films were evaluated using Discovery TGA 500 and DMA Q800 (TA, New Castle, DE, USA). The thermogravimetric analysis (TGA) of each sample (5–10 mg) was carried out in the temperature range 50–700 °C, and the heating rate was set as 10 °C min^−1^ under N_2_ purging (30 mL min^−1^). The dynamic thermomechanical properties of the nanocomposites were analyzed using a heating rate of 3 °C min^−1^ in the temperature range −50 to 100 °C.

### 2.9. UV-Vis Spectra

The visible light transmittance of the PVA/CNF films was determined in the wavelength range from 200 to 800 nm by using a PerkinElmer Lambda 950 UV-visible spectrophotometer, (Waltham, MA, USA).

### 2.10. The Degree of Carboxylation of TEMPO-Oxidized CNFs

The degree of carboxylation of TEMPO-oxidized CNFs was investigated by electric conductivity titration method [22,46]. Herein, 55.3 g CNF suspensions (containing 0.3 g CNF) and 0.01 M NaCl solution (5 mL) were mixed and stirred sufficiently to prepare a well-dispersed slurry. Then the above mixture was acidified with 0.1 M HCl solution and stirred again for 1 h at a pH value in the range of 2.5–3.0, thus ensuring all the carboxylate(−COO^−^) to be changed to carboxyl groups (−COOH) for further measurement. Subsequently, 0.04 M NaOH solution was dropwise added at the rate of 0.1 mL/min until pH = 11. During titration, conductivity meter was used to monitor the change of conductivity of solution in real-time. Finally, the titration curve of the NaOH consumption and conductivity was plotted, and the carboxylate content of the sample was determined from the titration curves. The content of carboxyl groups (*C*) was determined using the following equation:(1)$$C=\frac{\left(V_{2}-V_{1}\right)\times C_{NaOH}}{m}$$
where *V_1_* and *V_2_* are the volume of the used NaOH solution (mL), corresponding to the intersection point of the linear fitting lines between the first and second stage and between the second and third stage, respectively. *C_NaOH_* represents the concentration of the NaOH solution (0.04 M), and *m* refers to the weight of the CNFs.

## 3. Results and Discussion

### 3.1. Characterization of CNF with Different Lengths

Nano-CNF with different lengths were fabricated by the TEMPO catalytic oxidation. The preparation method is a simple process and does not require the complex and expensive equipment. As observed from the TEM images of CNF20 (Figure 2a), CNF10 (Figure 2b) and CNF5 (Figure 2c), the diameters of the CNFs are in the nanoscale range, with the length successively increasing from a few hundred nanometers to a few microns. Figure 2d respectively shows the optical images of the CNF films. As observed, the color of the different CNF films exhibits a significant distinction owing to the dimensional changes in CNFs.

During the TEMPO catalytic oxidation, NaClO reacts continuously with the jute fibers as an oxidant. Therefore, under certain pH conditions, the content of NaClO directly determines the degree of CNF reaction. The greater the amount of NaClO, the more hydroxyl groups on C6 in the cellulose are oxidized. As a result, the TEMPO oxidation becomes more thorough, thus, reducing the length. Thus, the nanocellulose with different morphology and size can be designed by adjusting the addition of NaClO.

CNF20 presents a uniform structure with a diameter of 5–10 nm and a length of several hundred nanometers to a few microns, thus, exhibiting a resultant aspect ratio of more than 50. Also, CNF10 exhibits a relatively uniform size under 200 nm, which is larger in length as compared to CNF20. In addition, entanglements among the fibers are observed, which may be caused by the fact that a small amount of CNF are not completely peeled off and are still connected to each other. As noted from Figure 2c, the size of CNF5 is much larger than the other two, with the largest diameter up to tens of nanometers. The observed phenomenon may be due to the low concentration of NaClO leading to a lower degree of oxidation, thus, a majority of CNF are not been stripped out, thus, leading to a significant extent of non-nanoscale cellulose in the system.

The observed phenomenon was further confirmed by optical images of the films prepared using different nanocellulose materials. As can be observed from the Figure 2d, on increasing the NaClO dosage, the transparency of the CNF membrane is enhanced. For a low NaClO content, the size of the cellulose fibers is in the micron range (or even visible to the naked eye), thus, the macroscopic materials do not demonstrate an effective light transmittance. On increasing the proportion of NaClO, the size of the cellulose fibrils reaches the nanoscale, and the CNF are closely packed with each other, thereby imparting optimal optical properties to the films.

As per the previous studies, changing the dosage of sodium hypochlorite does not affect the crystalline transformation of cellulose, with only change observed in the morphology and size of nanocellulose [47].

The characterization of the chemical structure, crystal structure and tensile properties of the different CNF materials are summarized in Figure 2e–f. As observed from the infrared spectra in Figure 2e, the nanocellulose materials exhibit the characteristic absorption peaks at 3340, 2901, 1595, 1105, 898 cm^−1^ with slight differences in intensities among different samples [48,49]. It indicates that the chemical structure of CNF largely remains the same. Specifically, a strong absorption peak around 3300 cm^−1^ is observed, corresponding to the stretching vibration of the hydroxyl group [50]. The intensity of the O–H stretching absorption peak was more pronounced for the CNF20 and CNF10 compared with CNF5, showing a higher proportion of cellulose within fibers [51]. The vibration peak at 2910 cm^−1^ is assigned to the C–H stretching vibration of the methylene group, whereas the C–H symmetric bending of CH_2_ appears at 1410 cm^−1^ [52,53]. At the same time, CNF exhibit characteristic bonds at 1595, 1105 and 898 cm^−1^, which are attributed to the stretching vibration of COONa derived by the TEMPO-mediated oxidization, C–O–C stretching of the ether bond and characteristic absorption of the cellulosic anomeric carbon C1, respectively [46,47,48,49,50,51]. Moreover, the intensity of the COO^−^ peak exhibits the following order: CNF20 > CNF10 > CNF5, indicating the degree of carboxylation increased with increasing NaClO contents, which consists with conclusion of Puangsin et al [51,54].

Figure 2f presents the 1D integrated SR-WAXS curves of CNF20, CNF10 and CNF5. The integral curves display two strong diffraction peaks at 2θ = 18.1° and 27.8°, corresponding to the lattice planes of cellulose ((200) and (004)). Moreover, two overlapping weak diffraction peaks at 2θ = 11.5° and 13.5° can be attributed to the Bragg peaks of cellulose ((110) and ($1\bar{1}0$)). Overall, the different nanocellulose materials with various dimensions do not exhibit any changes in the crystalline structure.

The zeta potential is an important index representing the stability of a colloidal system, which manifests the degree of repulsion of the adjacent particles with the same charge in the dispersion. The higher is the absolute value of the zeta potential, the better is the stability of the colloidal dispersion. In other words, the ability to dissolve or disperse is higher than that of agglomeration [55]. The zeta potential distribution of CNF20, CNF10 and CNF5 is illustrated in Figure 2g. As observed, CNF20, CNF10, and CNF5 possess negative ξ values, peaking at −62.8, −54.7 and −37.2 mV, respectively. The absolute value of the zeta potential of nanocellulose is noted to increase with the NaClO addition, thus, indicating that the CNF dispersion becomes more electrically stable. It is due to the reason that a higher content of NaClO promotes the oxidation, so that a large number of primary hydroxyl groups on the surface of CNF are converted into COO^−^, and a strong electrostatic repulsion between the microfibers in water contributes to the uniform dispersion of CNF in the aqueous solution. The observed finding is consistent with the TEM observations.

The mechanical behavior of the CNF films is presented in Figure 2h. As observed from the stress-strain curve, CNF10 exhibits the best mechanical properties, with the tensile strength and Young’s modulus reaching up to 126.38 MPa and 7.4 GPa. In contrast, the tensile strength of CNF20 is 101.58 MPa, which is slightly lower than CNF10. On the other hand, its modulus is about 1.1 GPa lower than CNF10 (7.4 GPa). It is worth noting that the modulus of CNF5 is as low as 2.14 GPa, and the tensile strength is much lower as compared to CNF20 or CNF10, which may be attributed to the inhomogeneous dispersion of cellulose in the PVA matrix. In general, the CNF with different sizes exhibit completely different mechanical properties, which is important for exploring the different behaviors of the reinforced composites.

### 3.2. Determination of the Content of Carboxyl Groups and Carboxylate in CNFs

Figure 3a–c shows the conductivity titration curves of CNFs with different dimensions. As is shown in Figure 3a–c, the relationship of the conductivity with the consumption of sodium hydroxide solution can be divided into three stages. With the increasing volume of sodium hydroxide, the conductivity decreases linearly in the first stage, changes slowly and tends to plateau in the second stage, and then begins to increase linearly in the third stage. The consumption of sodium hydroxide solution is directly relevant to the original carboxyl groups and the newly generated carboxyl groups from carboxylate after HCl acidification. According to the above Equation (1), the total content of carboxyl groups and carboxylate in CNFs can be calculated as 0.77 mmol g^−1^, 0.75 mmol g^−1^ and 0.34 mmol g^−1^, respectively, which further demonstrates that more NaClO content promotes TEMPO-mediated oxidation, and consequently increases the degree of carboxylation of CNF.

### 3.3. Thermal Properties of CNFs with Different Lengths

The thermogravimetric analysis (TGA) and derivative thermogravimetric (DTG) curves of CNFs are exhibited in Figure 3d–e. The degradation temperature (T_onset_) and maximum degradation temperature (T_max_) are listed in Table 1. The onset thermal degradation temperature of TEMPO-oxidized CNFs is 206.93, 220.73, 225.21 °C, respectively, which is lower than jute fiber (>250 °C). The lower degradation temperature of TEMPO-oxidized CNFs is attributed to decarboxylation of sodium anhydroglucuronate units and degradation of cellulose [51]. At the same time, the two main sharp peaks could be clearly observed from the DTG curves of CNFs, which represents the fastest degradation rate. According to Table 1, the T_max1_ and T_max2_ of CNF5 are all higher than CNF10 and CNF20. Besides, a higher content of residues at 800 °C (>25.0%) is observed for three kinds of CNFs, which is induced by a sodium cation acting as a counterion [56]. In conclusion, with the increase of sodium hypochlorite dosage, the carboxyl contents on cellulose surfaces increase and the decreased thermal stability connected with a higher residue content could be obtained [57].

### 3.4. The Effects of Morphology and Dimension of CNF on PVA/CNF Composites

#### 3.4.1. Chemical Strcture

The differences in the chemical structure of PVA and CNF can be evaluated from the infrared spectra in Figure 4a. Both PVA and CNF exhibit strong absorption peaks around 3300 cm^−1^, which represents the stretching vibration of the hydroxyl group [58]. The absorption bonds at 2910 cm^−1^, 1410 cm^−1^ and 1050 cm^−1^ are assigned to the C–H stretching vibration of the methylene base, C–H bending vibration peak of methylene and C–O stretching, respectively [59]. On the other hand, the characteristic absorption peaks of CNF are observed at 1595, 1105 and 898 cm^−1^ [53], which are also consistent with Figure 2e. However, in the FTIR spectra of the PVA/CNF composite films, the characteristic absorption peaks of CNF are not obvious, owing to the small content of CNF.

#### 3.4.2. Optical Transparency

Owing to the superior optical transparency of the neat PVA films, the morphology of nanocellulose and its distribution in the PVA matrix play a dominant role in the visible light transmission of the nanocomposites [25]. The UV-vis spectra of the PVA/CNF nanocomposite films are presented in Figure 4c–e. The transmittance (Tr., %) of pure PVA film is noted to be as high as 90%, and the composite films also exhibit a remarkable optical transparency. Compared with the PVA/CNF10 and PVA/CNF5 composites, the PVA/CNF20 films display the highest optical transmittance for the same CNF concentration. The Tr. values of PVA/CNF20–0.5%, PVA/CNF10–0.5% and PVA/CNF5–0.5% are determined to be 87.80%,87.75% and 87.42%, respectively, which are only about 3% lower than the transmittance of the pure PVA film (90.84%). In addition, as also observed from the Figure 4c–e, the Tr. of PVA/CNF20–2% and PVA/CNF10–2% decreases to 83.51% and 80.51%, respectively, while the light transmission of PVA/CNF5–2% shows a remarkable decline to 77.55%, thus, further indicating the effect of the uniform size of CNF20 as well as its perfect dispersion and distribution in the PVA matrix.

#### 3.4.3. Mechanical Properties

Figure 4f–h and Table 2 present the tensile modulus, tensile strength and elongation-at-break of the composite materials. The typical stress-strain curves of the samples are also presented in Appendix A.

Compared to the raw PVA film, the Young’s modulus of the PVA/CNF composite films display a significant increase as a function of the CNF content. Particularly, the tensile modulus of the PVA/CNF10 composites is observed to increase by 430% after the addition of 2% CNF10. A similar behavior is also observed for tensile strength, where a rising trend is noticed as a function of the nanocellulose loading. However, as the CNF content reaches 2%, the tensile strength is observed to decrease slightly, probably owing to the poor distribution caused by the agglomeration of nanocellulose. Although a similar trend is noticed for the different nanocellulose materials, not all CNF materials are observed to generate a uniform reinforcement of the polyvinyl alcohol matrix. The incorporation of CNF5 results in a gradual decline in the tensile strength of the PVA composite film. The possible reason is that the presence of the non-nanoscale cellulose in CNF5 hinders its homogeneous dispersion in PVA and weakens the hydrogen bond interaction between CNF and PVA; thus, the polymer molecular chains cannot be compactly stacked, and the defects are easily formed. Besides, the elongation at break suffers a successive reduction on increasing the CNF content for all PVA/CNF nanocomposite films, which is attributed to the tough network structure formed between the cellulose nanofibirls, thus, thoroughly restricting the polymer chain activity [22,34].

#### 3.4.4. Thermal Properties

The effect of CNF on the thermodynamic behavior of the PVA/CNF nanocomposites was characterized using thermogravimetric analysis (Figure 5). As observed, the samples exhibit a two-step thermal degradation, which is consistent with the degradation of PVA, as reported previously [60]. First, the dehydration reaction of PVA occurs at 220–350 °C, as shown in the following equation:
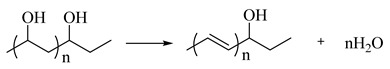
(2)

Second, the polyene residues degrade to form a mixture of carbon and hydrocarbons. The typical temperatures at 5% weight loss (T_−5%_) and 25% weight loss (T_25%_) are used to study the thermal stability of the PVA composite films, as manifested in Table 3. T_−5%_ of PVA is noted to be approximately 263 °C. Incorporating CNF (especially CNF20) in the PVA matrix shifts the T_−5%_ and T_25%_ values toward higher temperatures, i.e., the thermal stability of the PVA films is improved on incorporating CNF20 and CNF10. Particularly, only 1% filler loading of CNF20 leads to a remarkable increase of 25 °C in T_25%_, which can be attributed to the hydrogen bonds derived from CNF and PVA. The hydrogen bonds inhibit the movement of the polymer molecular chains, thereby forming a denser structure in the nanocomposites, which restricts the degradation of the PVA matrix [22,34,38]. However, the thermal stability of PVA/CNF5 is noted to decrease, as the thermal degradation of the non-nanoscale cellulose material accelerates the thermal decomposition of PVA [61,62,63,64].

**Figure 5 polymers-14-00128-f005:**
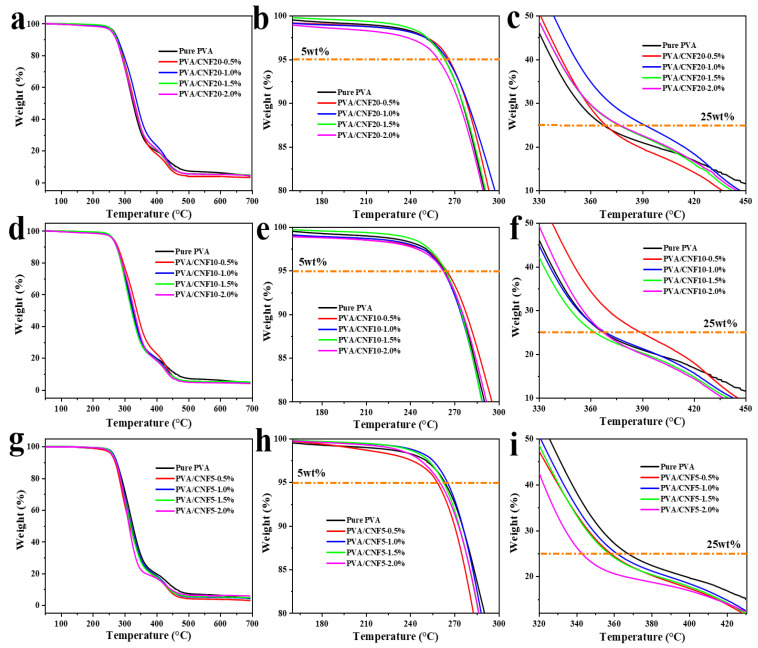
The TGA curves of (**a**–**c**) PVA/CNF20; (**d**–**f**) PVA/CNF10 and (**g**–**i**) PVA/CNF5 composites at a heating rate of 10 °C min^−1^.

The high strength, dimensional anisotropy and biocompatibility of CNF make them a functional and renewable reinforcing agent for the polymer composites. The compatibility between the matrix and filler as well as the interfacial effect directly determine the composite properties. If the filler is not uniformly dispersed in the matrix or the interface adhesion with the matrix is weak, the mechanical properties of the composites cannot be enhanced or may even deteriorate. Thus, it is necessary to analyze the mechanical properties for exploring the affinity between the reinforcing agent and matrix as well as the relationship between the macro-performance and microstructure of the material [21]. The mechanical reinforcing effect of CNF in the PVA matrix was evaluated by using the conventional electronic universal testing and dynamic mechanical analysis (DMA).

Figure 6 and Table 4 exhibits the evolution of the storage modulus and tan δ versus temperature for the PVA/CNF nanocomposite films as a function of the nanocellulose size and proportion (i.e., 0, 0.5, 1.0, 1.5 and 2 wt%). As observed from the DMA curves and DMA data, a small extent (less than 1 wt%) of CNF can make the storage modulus of the composite about 5−12%. Once the CNF content reaches or even exceeds 1%, a steady improvement in the viscoelasticity of the obtained nanocomposites is obtained, especially above the glass transition temperature (Tg). Specifically, PVA/ CNF5 showed the highest storage modulus (1453 MPa) when the CNF dosage was 1.0%, while PVA/CNF20 exhibited 53.4% higher storage modulus than pure PVA when the dosage was 1.5%. The CNF addition of 2.0% resulted in slightly lower storage modulus than that of 1.5%. The observed enhancement trend might be associated with the restricted network structure formed by CNF in the PVA matrix [25].

#### 3.4.5. The Comparison of this Work with Other Researches

The results of this work compared with other researches were summarized in Table 5. Compared PVA/MFC, the PVA/CNF films in this work possess lower tensile strength and lower Young’s modulus, which may be attributed to the smaller diameter and lesser CNF content. However, the PVA/CNF composites exhibit higher tensile strength and higher Young’s modulus contrast with EVA/CNF when the added CNF have similar size. In addition, the added CNF is CNF-I while the strength and modulus are lower than PVA/CNF-I, which can be caused by filming technology. In general, the PVA/CNF films in this work show relatively superior mechanical property, optical transparency and thermodynamic property, which will make them have a wide application prospect in the future.

## 4. Conclusions

In summary, a novel, simple and inexpensive strategy has been demonstrated for preparing the high performance polymer nanocomposites with tunable mechanical properties, based on the reinforcement of the PVA matrix with abundant and renewable CNF. Three kinds of CNF with different dimensions were produced by using the classical TEMPO catalytic oxidation by altering the NaClO content. The CNF grades exhibited distinct chemical structure, crystalline structure, light transmission, gel stability and mechanical properties. Subsequently, the reinforcement effect of the different CNF on the optical transparency, thermostability and mechanical properties of the PVA nanocomposites was thoroughly investigated. First, despite a decline in the optical transparency caused by CNF, the PVA/CNF films demonstrated a favorable visible light transmittance (Tr. > 75%). Meanwhile, the inherent nature of CNF, homogeneous dispersion and compatibility between PVA and CNF also had a significant impact on the mechanical properties and thermal stability of the PVA/CNF nanocomposite films. Due to the high strength and modulus of CNF10, the PVA/CNF films demonstrated a superior Young’s modulus of 482.75 MPa at 2 wt% filler loading. Further, the presence of CNF may induced the nucleation of the PVA matrix, which further contributed to the generation of the steady and robust 3D network.

## Figures and Tables

**Figure 1 polymers-14-00128-f001:**
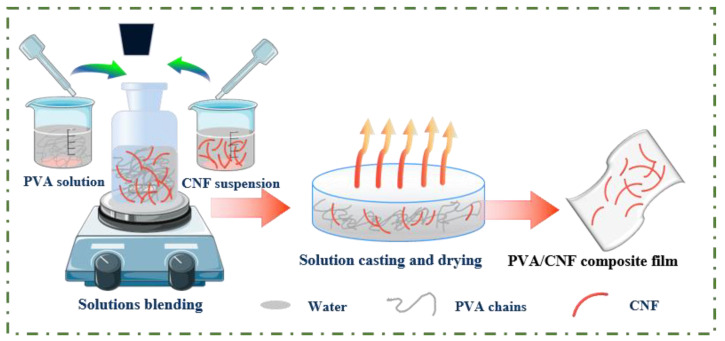
The schematic representation of the synthesis of the PVA/CNF nanocomposite films.

**Figure 2 polymers-14-00128-f002:**
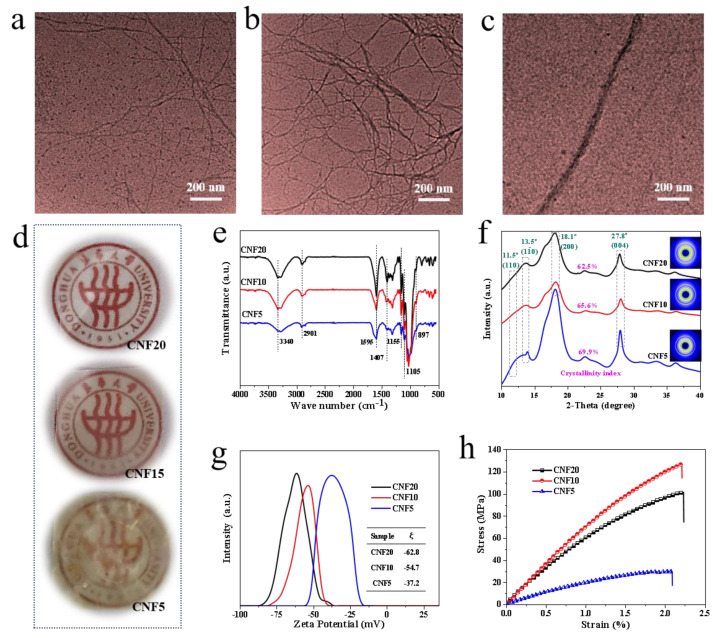
The TEM images: (**a**) CNF20; (**b**) CNF10; (**c**) CNF5; (**d**) Optical images of the films; (**e**) FTIR spectrum of CNF/PVA; (**f**) 1D-WAXS of CNF/PVA; (**g**) Zeta potential distribution of CNF/PVA; (**h**) Stress-strain curves of CNF/PVA.

**Figure 3 polymers-14-00128-f003:**
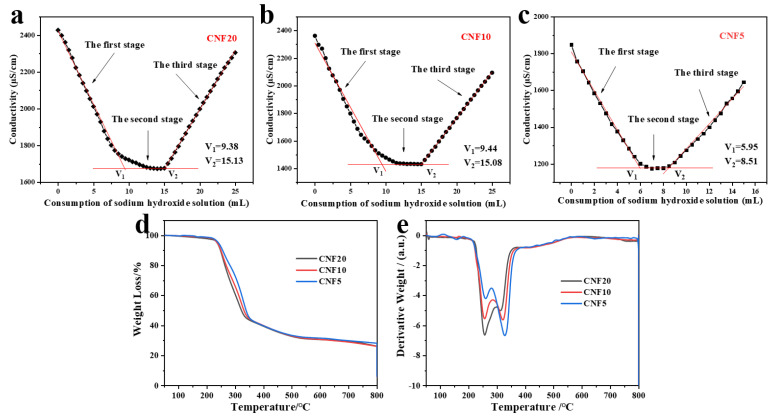
The conductivity titration curves (**a**–**c**), TGA curves (**d**), and DTG curves (**e**) of CNFs with different dimensions.

**Figure 4 polymers-14-00128-f004:**
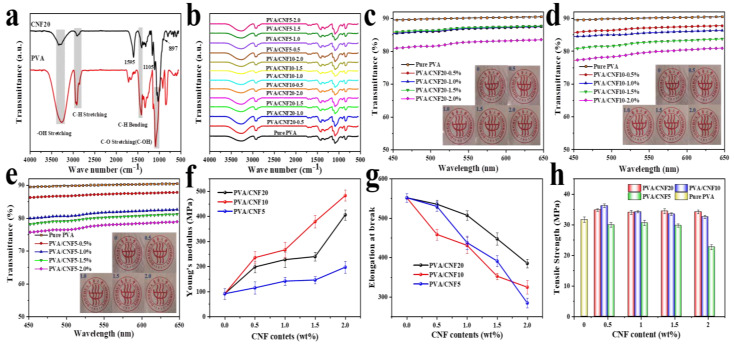
FTIR: (**a**) CNF20 and PVA films; (**b**) PVA/CNF composites with different additions of CNFs with different lengths; UV-vis spectrum of PVA/CNF composites: (**c**)PVA/CNF20;(**d**) PVA/CNF10; (**e**) PVA/CNF5; Mechanical properties of PVA/CNF composites: (**f**) Young’s Modulus; (**g**) Elongation at break; (**h**) Tensile strength.

**Figure 6 polymers-14-00128-f006:**
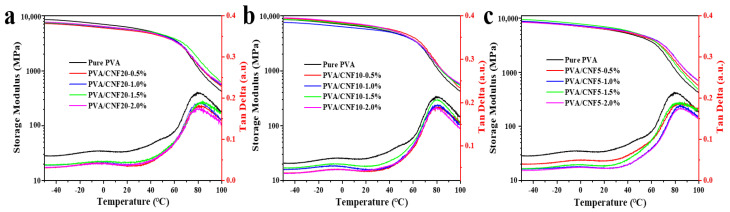
The DMA thermograms of neat PVA and PVA/CNF nanocomposites as a function of temperature: (**a**) PVA/CNF20; (**b**) PVA/CNF10; (**c**) PVA/CNF5.

**Table 1 polymers-14-00128-t001:** TGA data of CNFs with different morphology.

Samples	T_onset_/°C	T_max1_/°C	T_max2_/°C	Char Yield/%
CNF20	206.93	256.08	313.85	25.90
CNF10	220.73	257.29	321.44	25.89
CNF5	225.21	259.36	327.99	28.06

**Table 2 polymers-14-00128-t002:** The tensile properties of pure PVA and PVA/CNF nanocomposites.

Film	CNF Content (%)	Tensile Strength (MPa)	Elongation at Break (%)	Young’s Modulus (MPa)
PVA	0	31.65 ± 0.85	550.61 ± 11.06	91.00 ± 21.64
PVA/CNF20	0.5	34.84 ± 0.44	535.26 ± 10.71	197.73 ± 22.03
	1.0	34.02 ± 0.71	507.12 ± 11.65	227.41 ± 30.59
	1.5	34.46 ± 0.80	447.24 ± 15.49	239.29 ± 17.22
	2.0	34.22 ± 0.66	385.04 ± 10.73	405.80 ± 20.34
PVA/CNF10	0.5	36.21 ± 0.62	458.49 ± 13.85	234.98 ± 24.25
	1.0	34.21 ± 0.33	430.71 ± 20.47	266.22 ± 30.66
	1.5	33.48 ± 0.43	352.42 ± 8.58	378.88 ± 25.25
	2.0	32.55 ± 0.56	324.52 ± 6.78	482.75 ± 21.95
PVA/CNF5	0.5	29.97 ± 0.72	529.64 ± 11.06	114.48 ± 24.21
	1.0	30.64 ± 0.85	437.54 ± 11.75	141.27 ± 16.57
	1.5	29.75 ± 0.48	390.89 ± 17.21	145.77 ± 14.87
	2.0	22.71 ± 0.82	284.42 ± 12.48	196.83 ± 23.33

**Table 3 polymers-14-00128-t003:** The TGA data of neat PVA and PVA/CNF composites as a function of the CNF content.

Film	CNF Content (%)	T−_5%_/°C	T_25%_/°C
PVA	0	263.07	367.21
PVA/CNF20	0.5	265.97	368.63
	1.0	264.95	391.12
	1.5	262.94	376.18
	2.0	258.38	376.81
PVA/CNF10	0.5	264.51	388.55
	1.0	262.27	368.36
	1.5	263.84	361.96
	2.0	262.21	367.24
PVA/CNF5	0.5	258.25	358.17
	1.0	265.17	361.43
	1.5	262.56	357.07
	2.0	260.15	342.87

**Table 4 polymers-14-00128-t004:** The DMA data of PVA and PVA/CNF composites as a function of the CNF content.

Samples	PVA	20–0.5	20–1.0	20–1.5	20–2.0	10–0.5	10–1.0	10–1.5	10–2.0	5–0.5	5–1.0	5–1.5	5–2.0
Tg/°C	80	80	80	80	80	80	80	80	80	83	82	85	84
ΔE’ at Tg /MPa	1096	1225	1289	1681	1333	1156	1376	1173	1142	1263	1453	1030	1370

**Table 5 polymers-14-00128-t005:** The results of this work compared with other researches.

Samples	CNF Morphology		σ/MPa		ε/%		Ε/MPa		Tr./%		T_d_/°C		References
	D/nm	L/nm	After	Before	After	Before	After	Before	After	Before	After	Before	
PVA/MFC	365	-	48 ^1^	41	-	-	1600 ^1^	1200	-	-	265^1^	252	[65]
EVA/CNF	5–10	-	3.64 ^2^	3.32	688 ^2^	750	6.92 ^2^	5.36	77 ^2^	78	-	320–380	[51]
PVA/CNF-Ⅰ	10–15	1120	44.30 ^3^	39.08	89.2 ^3^	320.5	1473.86 ^3^	96.09	53 ^3^	90	294.9 ^4^	272.5	[25]
PVA/CNF-Ⅱ		250	44.59 ^3^	39.08	313 ^3^	320.5	276.2 ^3^	96.09	86 ^3^		287.4 ^4^	272.5	
PVA/CNF20	5–10	200–1000	34.22 ^5^	31.65	385.04 ^5^	550.61	405.80 ^5^	91	83.51 ^5^	90.84	258.38 ^5^	263.07	This work
PVA/CNF10	10–15	1000–3000	32.55 ^5^		324.52 ^5^		482.75 ^5^		80.51 ^5^		262.21 ^5^		
PVA/CNF5	20–50	>3000	22.71 ^5^		284.42 ^5^		196.83 ^5^		77.55 ^5^		260.15 ^5^		

Abbreviations: σ = Tensile Strength; ε = Elongation at break; E = Young’s Modulus; Tr. = Transparency; T_d_ = Thermal degradation temperature; D = Diameter; L = Length. ^1^ CNF content was 3%; ^2^ CNF content was 1%; ^3^ CNF content was 10%; ^4^ CNF content was 5%; ^5^ CNF content was 2%.

## Data Availability

The data presented in this study are available upon request from the corresponding author.

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
