# Peer review of "Effect of Length of Cellulose Nanofibers on Mechanical Reinforcement of Polyvinyl Alcohol"

_polymers, 2021, doi:10.3390/polym14010128_

Round 1

Reviewer 1 Report

This is an interesting article where authors have demonstrated a novel, simple and inexpensive strategy for preparing the high-performance polymer nanocomposites with tunable mechanical properties, based on the reinforcement of the PVA matrix with abundant and renewable CNF. 
Three kinds of CNF with different dimensions were produced by using the classical TEMPO catalytic oxidation by altering the NaClO content. The CNF grades exhibited distinct chemical structure, crystalline structure, light transmission, gel stability and mechanical properties. Following corrections are needed:

-Preparation of CNF of different sizes should be explained in more detail and authors should demonstrate how the work is different from others

-Abstract should display the quantitative results.

-Remove the background from figure 1

-The intext in figure 5 is small. please enhance the quality

-How the uniformity of the structure was maintained

-The FTIR results should be explained in more detail

-Have the authors studied other loadings as well

-Authors may consider citing relevant articles such as: International Journal of Biological Macromolecules 182, 1554-1581(2021); Industrial Crops and Products 170, 113780 (2021); Polymers 12 (7), 1472 (2020);  Polymers 2021, 13(23), 4060;

Reviewer 2 Report

Comments to authors are listed below:

  • Authors should report the significant finding with numerical values in the abstract section.
  • In the last paragraph of the Introduction section, the authors should state the novelty and applications of this study.
  • The characterizations and applications of PVA should be reported clearly to improve the quality of the introductions. So, some references are recommended to include in the introduction section:
  1. https://link.springer.com/article/10.1007/s42452-019-1111-2.
  2. https://www.degruyter.com/document/doi/10.1515/ipp-2020-3974/html.
  3. https://dergipark.org.tr/en/pub/jotcsa/issue/60129/878495.
  4. https://link.springer.com/article/10.1007/s10924-019-01470-7.
  5. https://www.sciencedirect.com/science/article/pii/S0144861718302571.

Reviewer 3 Report

Dear authors,

The topic is very interesting and well presented. The effect of the CNF length on mechanical properties has been rarely studied, therefore the present study presents contribution to this field of research.

Nevertheless, corrections should be made and here are my comments:

General comments:

  • Correct the figures, marked with red in the manuscript.
  • Number the chapters and sub chapters. Only introduction is numbered with 1….

1) Introduction

It provides sufficient background and includes all relevant references. The references are cited correctly. The aim and the goal of the research are clearly presented.

2) Materials

What were the properties of the jute fibres: length, diameter, maybe other specification from  the producer?

Line 86-94: add, where missing, country/state of the used materials.

3) Methods

Preparation of PVA/CNF films: line 122: what was the humidity at drying (since it was performed in room temperature)?

FTIR: line 133: add name of the producer and country of the FTIR instrument.

Mechanical properties: line 141: add the producer and country of the Instron testing machine.

Line 155: The same at UV-VIS spectra: add the producer and country.

4) Results and discussion

Infrared spectra: Line 240 – 248: add literature to corresponding defined peaks.

5) Conclusion

The conclusion is well presented. All important findings are included in this chapter.

6) References: Are cited correctly.  

Reviewer 4 Report

Cellulose nanofibers (CNFs) prepared by TEMPO-mediated oxidation with various contents of NaClO were added into polyvinyl alcohol to prepare composites. Discussion based on CNFs with different morphologies should be provided. Other comments should be carefully addressed in the context prior to consideration.

-The experimental part should be modified to show the composite preparation method. So, other researchers could follow the steps.

-Degree of carboxylation of each CNF should be measured and presented

- CNFs treated with various NaOCl contents presented lengths in several microns, but a change in fiber diameters was found. The diameter contribution of CNF materials should be presented. Also, the title of this work should be modified to effect of diameters of cellulose nanofibers on mechanical reinforcement of PVA.

-Mechanical properties of CNF-PVA composites presented line 228-237 should be removed as this part was mainly presented between line313 -333. Also, what is different between these two parts? Values presented were too much different. This should be carefully checked.

-FTIR, transparency and mechanical parts should be merged altogether to decrease redundancy.

-TGA and DTG curves of each CNF should be presented to see effect of chemical treatment.

-No discussion could be found in transparency and DMA results. Please provide the discussion in this part to differentiate effects of CNFs.

- The addition of CNFs in PVA composites could not improve tensile strength. Please discuss this issue.

-Results of the following articles should be compared with those of this work.

https://www.tandfonline.com/doi/abs/10.1080/15440478.2021.1993483

https://pubs.rsc.org/en/content/articlelanding/2016/ra/c6ra14517e

https://www.sciencedirect.com/science/article/abs/pii/S1359835X14001936

Round 2

Reviewer 1 Report

Accept

Reviewer 4 Report

The manuscript has been modified, and should be ready for publication.